# Measurements and Modeling of Optical Turbulence in the Coastal Environment

Sasha Barnett [1], Joseph Blau [1], Paul Frederickson [2] and Keith Cohn [1,*]

[1] Department of Physics, Naval Postgraduate School, Monterey, CA 93943, USA; sasha.barnett@navy.mil (S.B.); blau@nps.edu (J.B.)
[2] Department of Meteorology, Naval Postgraduate School, Monterey, CA 93943, USA; pafreder@nps.edu
[*] Correspondence: krcohn@nps.edu

**Abstract:** The goal of this study was to characterize optical turbulence in the near-coastal environment. Measurements to obtain the refractive index structure parameter and other meteorological data were taken over the course of a month along the shore of Monterey Bay. The results were compared to a new version of the Navy Vertical Surface Layer Model (NAVSLaM), a model of turbulence originally developed for maritime environments but now extended to terrestrial environments. The new version has not been previously validated by comparisons to experiments, particularly in a complex environment such as near the coastline. Our experimental results showed generally good agreement between measured and modeled levels of turbulence. Specifically, the differences between experimental and modeled values of the refractive index structure parameter were less than an order of magnitude in most conditions and followed the same diurnal trend. There were some greater differences during near-neutral conditions, but this is a known limitation of the model. Overall, this extended model appears to do a good job of predicting turbulence in this environment for the observed time period.

**Keywords:** optical turbulence; atmospheric measurement; coastal environment; directed energy; free-space optical communications; atmospheric surface layer

## 1. Introduction

Optical turbulence can have significant effects on laser beam propagation, impacting the performance of directed energy weapons and free-space optical communication systems. Various models have been developed and numerous experiments conducted to characterize optical turbulence over the ocean and over land [1,2]. A particularly challenging environment to model is a near-coastal region, where turbulence can be impacted by the changing topography, heating of both the land and ocean surface, and the presence of on-shore or off-shore air flow [3,4].

The navy vertical surface layer model (NAVSLaM) was developed by the Naval Postgraduate School (NPS) Meteorology Department to model optical turbulence over the ocean. It produces profiles of the refractive index structure parameter, $C_n^2$, versus height in the surface layer (up to ~100 m) above the ocean surface. This model has been validated by comparisons to experiments [5,6]. A two-level version of NAVSLaM was recently developed to model turbulence over land. Both versions of NAVSLaM are based upon Monin–Obukhov similarity theory (MOST), which assumes that conditions are horizontally homogenous and stationary. It is unclear whether such a model is valid in a near-coastal environment, where surface and atmospheric conditions can change rapidly in both time and space. Further details on MOST and its application for modeling vertical profiles of $C_n^2$ can be found in Frederickson et al. [1].

The surface layer and other vertical profile models of turbulence can, in principle, be tested against radiosonde measurements. Radiosondes can measure turbulence profiles that extend many kilometers in altitude [7–9]; NAVSLaM, on the other hand, is applicable

only in the first tens of meters above the surface. Thus, to test the validity of the two-level version of NAVSLaM in a coastal environment, the NPS Physics Department recently conducted an experiment along the shore of the Monterey Bay using point measurements of turbulence just a few meters apart at two heights above the surface. This approach provided a near-continuous stream of data at a fixed location and probing heights very close to the surface, where the assumptions of NAVSLaM are presumably more plausible. Over approximately one month, we took measurements to obtain values of $C_n^2$ and other meteorological parameters, including air temperature, wind speed, and humidity, at two different heights. The meteorological data was then fed into the two-level version of NAVSLaM to predict $C_n^2$ values at the two heights, which were then compared to the measured values.

## 2. Materials and Methods

The original version of NAVSLaM is valid for applications only over the ocean, and it only requires atmospheric data at a single level above the surface. A newer version of NAVSLaM, which is used in this study, is valid over either land or ocean surfaces, and it requires inputs of measured or modeled atmospheric data (wind speed, air temperature, and humidity) at two height levels within the ASL. When using measured data as inputs to NAVSLaM, the values are normally averaged over a 5 to 30 min time interval.

The experiment took place on a coastal bluff overlooking the Monterey Bay from 16 March 2021 through 20 April 2021; data were collected nearly continuously throughout this timespan. The sensor platform itself, shown on the left in Figure 1, was placed on a coastal bluff about 70 m from the water's edge at high tide.

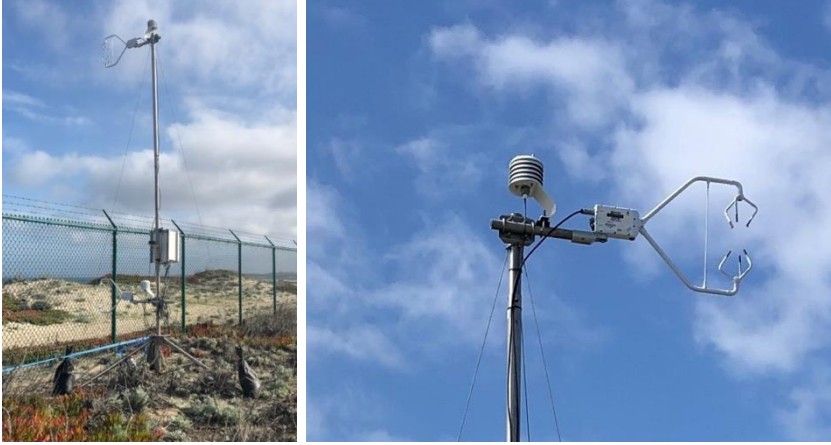

**Figure 1.** Sensor platform (**left**) and close-up of two of the sensors (**right**). The upper sensor is a HygroVUE5 hygrometer, and the one on the right is a CSAT3 sonic anemometer.

Since this new NAVSLaM model requires inputs of wind speed and air temperature collected at two different heights above the ground, the sensor tripod platform deployed an array of sensors clustered around two different heights. The base of the platform had one HygroVUE5 (Campbell Scientific) hygrometer fixed at 1.2 m above the surface with a CSAT3 sonic anemometer (Campbell Scientific) at 1.1 m. At the top of the platform, a second HygroVUE5 hygrometer and CSAT3 anemometer (both shown in Figure 1 on the right) were fixed at 4.4 and 4.2 m above the ground, respectively. Height measurements were taken from the center of the instrument to the surface. The hygrometers were enclosed in solar radiation shields, and a GPS unit was used for timestamps. All the data were recorded using a CR6 datalogger (Campbell Scientific); the CSAT3s were sampled at 50 Hz while the hygrometers recorded a sample every 5 s.

In addition to providing wind speed, the sonic temperatures recorded by the two CSAT3 anemometers were used for point estimates of the refractive index structure parameter $C_n^2$. These CSAT3-derived values of $C_n^2$ were then compared to the NAVSLaM

predictions at the respective heights of the sonics. This approach provided sufficient vertical resolution to discern the differences between $C_n^2$ at the measured heights in a robust, weatherized platform.

The conversion of the sonic anemometer data to $C_n^2$ is a multi-step process. First, the temperature structure-function, $C_T^2$, can be estimated from sonic temperatures measured using the CSAT3s. These data were split into tranches with 8192 samples each (corresponding to 164 s of collection time acquired at 50 Hz), then an estimated power spectral density (PSD) was obtained using Welch's method applied to each tranche. The relationship of the PSD $S_T$ and frequency-dependent temperature structure function $C_T^2$ is given by [5]:

$$C_T^2(f) = 4\left(\frac{2\pi}{U}\right)^{2/3} S_T(f)f^{5/3} \tag{1}$$

where $U$ is the mean wind speed as measured by the sonic anemometers for a particular tranche. This equation assumes that $f$ is a temporal frequency that corresponds to a spatial frequency within the inertial subrange. Hence, $S_T(f)$ plotted on a log–log plot should be linear with a characteristic $-5/3$ slope within temporal frequencies that correspond to the (spatial) inertial subrange; a linear fit to the log–log data with this slope was used to extract the value of $C_T^2$ for each tranche of data. For this experiment, a frequency span between 1 and 5 Hz was chosen to estimate $C_T^2$, as the PSD slope was near $-5/3$ throughout the collection period within this frequency window.

Turbulent cell sizes that are smaller than a sonic anemometer's transducer experience a path-averaging effect that attenuates the PSD spectrum at higher frequencies [10,11]. The frequency threshold above which this effect becomes prominent depends on the wind speed and the anemometer transducer geometry. For a given wind speed and sonic geometry, the transfer function between the "true" and path-averaged PSDs can be estimated and, therefore, this effect can be corrected. This was done following the procedure discussed in [12]. Finally, $C_n^2$ is estimated from $C_T^2$ using $C_n^2 \approx A^2 C_t^2$, where $A$ is a coefficient that depends on the optical wavelength, pressure, temperature, and humidity [5].

While sonic anemometers have a long track record for optical turbulence studies, they are by no means the only way to acquire point measurements of $C_n^2$. For example, pairs of calibrated fast-response thermometers (e.g., fine-wire thermocouples, optical temperature sensors) can measure the temperature structure function directly [13]. However, to permit rapid measurements, these devices rely on sensor elements with small thermal masses. The foggy environment around Monterey Bay often results in these elements being drenched with dew; this, in effect, adds thermal mass to the elements and severely limits their rapid response times. Sonic anemometers are more robust against this phenomenon.

Optical measurements using scintillometers or even image jitter can profile $C_n^2$, but these techniques all employ some sort of averaging or weighting along the path between the source and imaging optics [14,15]. For this work, we required point measurements of turbulence, so these optical techniques were inappropriate for this application.

## 3. Results

Data were collected from 16 March 2021 to 20 April 2021, as shown in Figure 2, where the overall $C_n^2$ values captured by the sonics and estimated by NAVSLaM are plotted. The $C_n^2$ values from the upper sonic anemometer are displayed in the first plot in red with the NAVSLaM estimates overlayed in blue. The $C_n^2$ values from the lower sonic anemometer are given in the second plot also in red with the NAVSLaM estimates overlayed in blue.

The final plot presents the mean air temperature difference (upper temperature minus lower temperature) as measured by the two hygrometers. The solid black line indicates near-neutral conditions when the mean temperature difference is zero. The gaps in the record are due to power interruptions; to avoid further interruptions, we connected solar panels to the power supply. Additionally, from the 20th of March to the 21st of March, the experimental setup was relocated to avoid interference with sensors monitored by the Oceanography Department and thus the reason for the gap in data for that period.

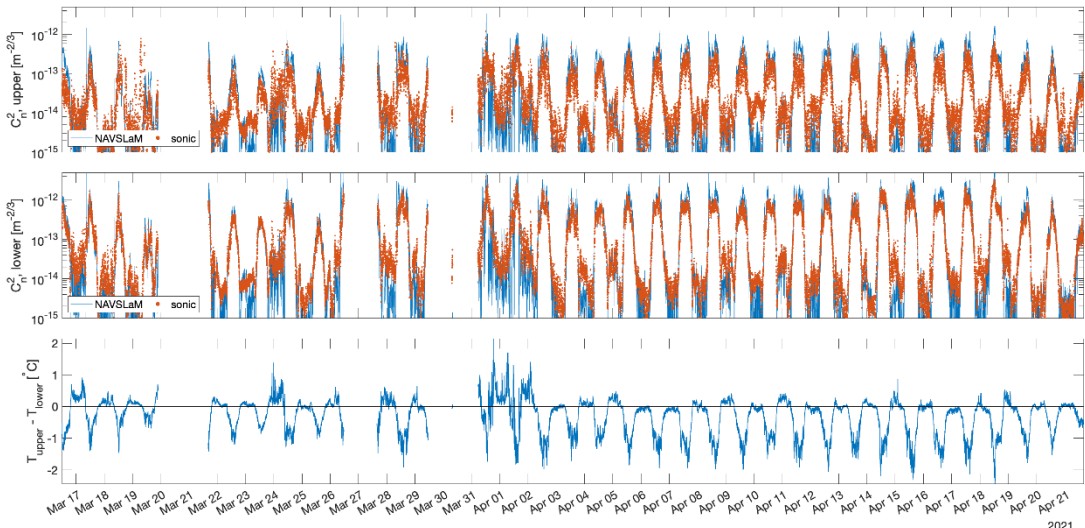

**Figure 2.** The upper two plots show overall $C_n^2$ estimates from NAVSLaM (blue curves) and CSAT3 sonic anemometers (red dots), for the entire period of data collection from 16 March 2021 to 20 April 2021. The bottom plot shows the mean air temperature difference between the upper and lower sonic locations.

Figure 2 shows that both estimates and measurements follow a typical diurnal pattern with the largest $C_n^2$ values occurring throughout the warmer afternoon hours and smaller $C_n^2$ values seen in the cooler evening hours into post-midnight hours. At the end of March, periods of time are shown when the temperatures at the higher elevation were warmer than temperatures collected closer to the surface. More importantly, Figure 2 shows generally good agreement between measured $C_n^2$ values and NAVSLaM estimates, especially during periods where the mean temperature difference is less than zero (i.e., during unstable conditions). However, periods of disagreement do exist and can be seen when the mean temperature difference is zero or, occasionally, greater than zero.

Generally, the temperature difference between the upper and lower locations resulted in unstable conditions during the day, as the air at the lower height was warmed due to its proximity to the ground. This temperature difference drove significant turbulence during these periods—the lower CSAT3 recorded values of $C_n^2$ more than $10^{-12}$ m$^{-2/3}$ by midday, while the values for the upper CSAT3 were somewhat less. The NAVSLaM predictions followed a similar trend. At nighttime, near-neutral conditions generally prevailed, resulting in much lower $C_n^2$ values recorded by the CSAT3s (as predicted by NAVSLaM).

However, there were rare occasions where this overall trend was not followed. Usually, the wind direction along the Monterey coastline is onshore. During offshore flow events, warmer inland air can mix with cooler maritime air, resulting in warmer air aloft. One such event occurred at the end of March, as shown in Figure 3. For example, unstable conditions exist when the wind is onshore, while the airmass becomes stable during periods of offshore flow. This trend was followed throughout the measurement campaign.

Figure 4 illustrates the results gathered over a 12-h period from 0600 Sunday, 4 April to 0600 Monday, 5 April, and is provided to demonstrate how the measured data were analyzed. The two upper plots display the $C_n^2$ values calculated from measurements recorded by the upper (blue curve) and the lower (orange curve) sonic anemometers using the process described in the preceding section. These values range from $10^{-16}$ m$^{-2/3}$ to $10^{-13}$ m$^{-2/3}$ and follow an expected daily trend in which the highest values occur in the middle of the day and the lowest values at morning and early evening hours as the temperatures transition through neutral conditions.

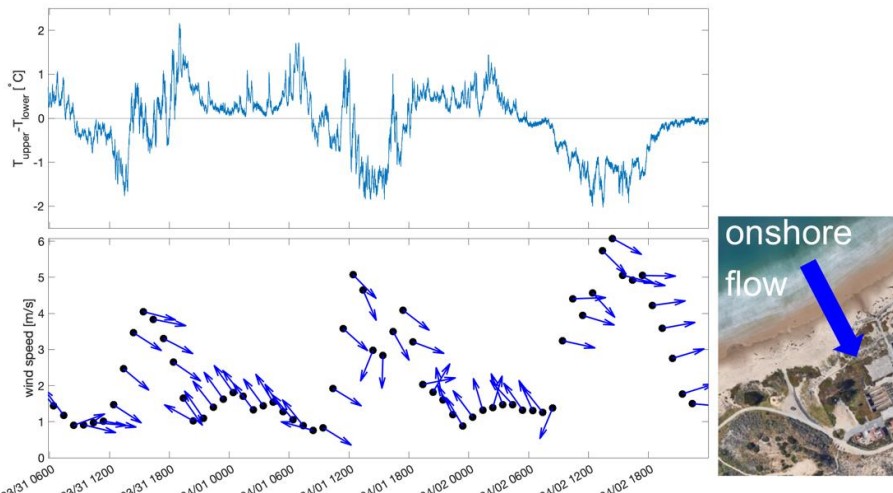

**Figure 3.** Periods of stable conditions (where the air temperature at the upper location was warmer than the lower location) were accompanied by offshore flow events. In the bottom diagram, the arrows correspond to the wind velocity direction. As show in the inset figure on the right, onshore flow occurs when the arrows point towards the lower right, whereas offshore flow occurs when the arrows point to the upper left.

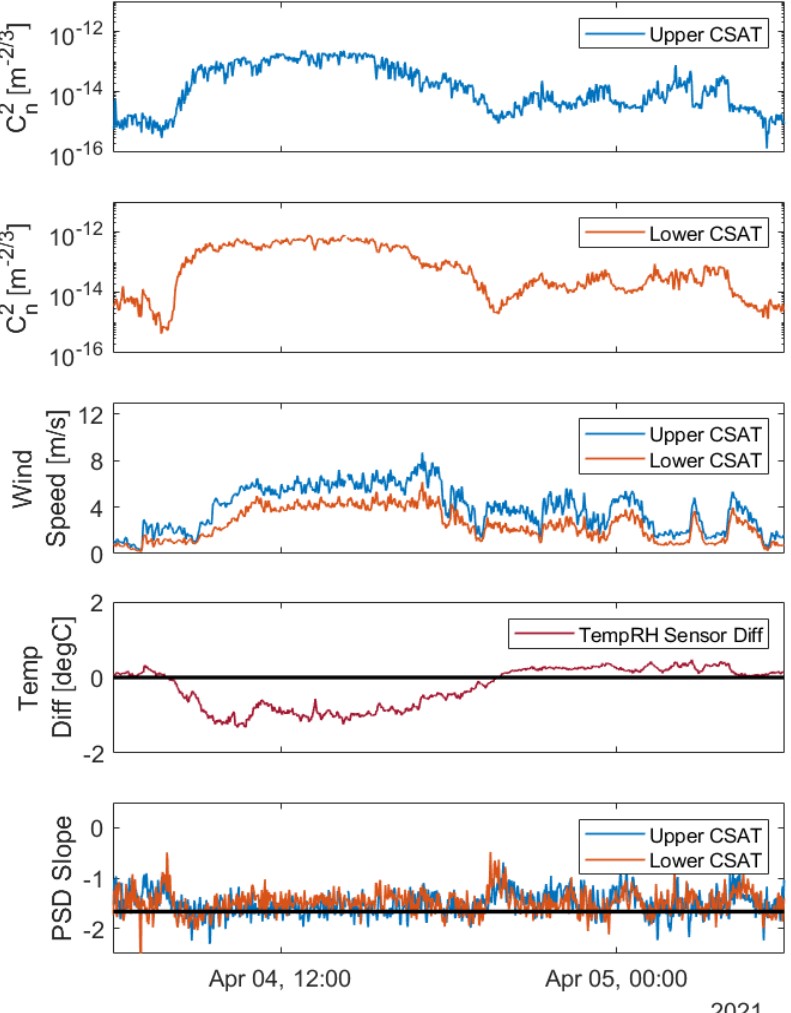

**Figure 4.** Data collected from 0600 Sunday, 4 April to 0600 Monday, 5 April (local time).

The fourth plot of Figure 4 is the temperature difference between the upper and lower hygrometers. Here we see an expected trend of the lower temperature readings being greater than the temperature measured by the upper sensor due to its proximity to the heat-absorbing surface. Overnight, we reach a point of inflection as the ground cools and is momentarily at equilibrium with the temperature of the air at the higher elevation, shown with the solid black line plotted. As we enter the post-midnight hours, the ground is now cooler than the air at the higher elevation. It is also important to note that—as expected—the $C_n^2$ values of both sonics tend to be higher during periods of large negative temperature differences and tend to be lower during neutral conditions (i.e., when the temperature difference is zero). Other instances such as these will be further explored when we compare our results to NAVSLaM estimates.

In the final plot of Figure 4, we have the slopes of the best line fit to the PSDs of both sonic anemometers; the expected value of $-5/3$ slope was plotted as a solid black line. This plot was used to ensure that the probing frequency window (1 to 5 Hz) was sufficient to capture frequencies within the inertial subrange, as previously discussed. In the early morning hours of April 4th, the slope deviates drastically from the expected value of $-5/3$; simultaneously, both sonic anemometers indicate a $C_n^2$ value of $\sim 10^{-16}$ m$^{-2/3}$. This may indicate that the sonics are approaching the noise floor which occurs when $C_n^2 \approx 10^{-16}$ m$^{-2/3}$; smaller values than this are difficult for these sonics to measure. We noted another strong deviation from the $-5/3$ slope just before 2000 on the night of the fourth. This also may be attributed to the sonics approaching their noise floors.

After analyzing the collected data from the sonic anemometers for the remaining days of the experiment, we then compared these results with estimates produced by the two-level version NAVSLaM. The inputs to NAVSLaM are listed in Table 1, along with the corresponding instrument used to determine that input. Wind speed, air temperature, and relative humidity were recorded at two different heights, as described earlier and as required by this version of NAVSLaM. The outputs of NAVSLaM include $C_n^2$ as a function of height above the surface. The sonic $C_n^2$ estimates were compared to the NAVSLaM outputs at the corresponding heights.

**Table 1.** Inputs to NAVSLaM for modeling $C_n^2$.

| Parameter | Input Source |
| --- | --- |
| Optical Wavelength | 1 µm |
| Wind Speed (upper) | Upper CSAT3 Sonic |
| Wind Speed (lower) | Lower CSAT3 Sonic |
| Temperature (upper) | Upper HygroVUE5 |
| Temperature (lower) | Lower HygroVUE5 |
| Relative Humidity (upper) | Upper HygroVUE5 |
| Relative Humidity (lower) | Lower HygroVUE5 |
| Pressure | 1000 hPa |
| Height of upper sensors | 4.4 m |
| Height of lower sensors | 1.2 m |

Figure 5 shows the $C_n^2$ estimates produced by NAVSLaM overlayed on plots from the sonics from 0700 Tuesday, 6 April to 0800 Wednesday, 7 April. Again, values measured by the two CSAT3 sonic anemometers are shown in blue (upper sonic) and orange (lower sonic). The green curves in the first two plots represent the NAVSLaM $C_n^2$ estimates at the corresponding heights of the sonics (4.4 m above the ground for the upper sonic and 1.2 m for the lower sonic). Additionally, the first two plots contain colored circles along the horizontal axis which represent various diagnostic errors recorded by the CR6 Datalogger. Most of the errors recorded were due to low battery warnings in which slow scans would be occasionally skipped; however, full data scans were still conducted and saved to the

CR6 Datalogger. While these errors did not pose issues with the overall analysis of the data, they were still recorded as a flag to ensure those data points received additional attention.

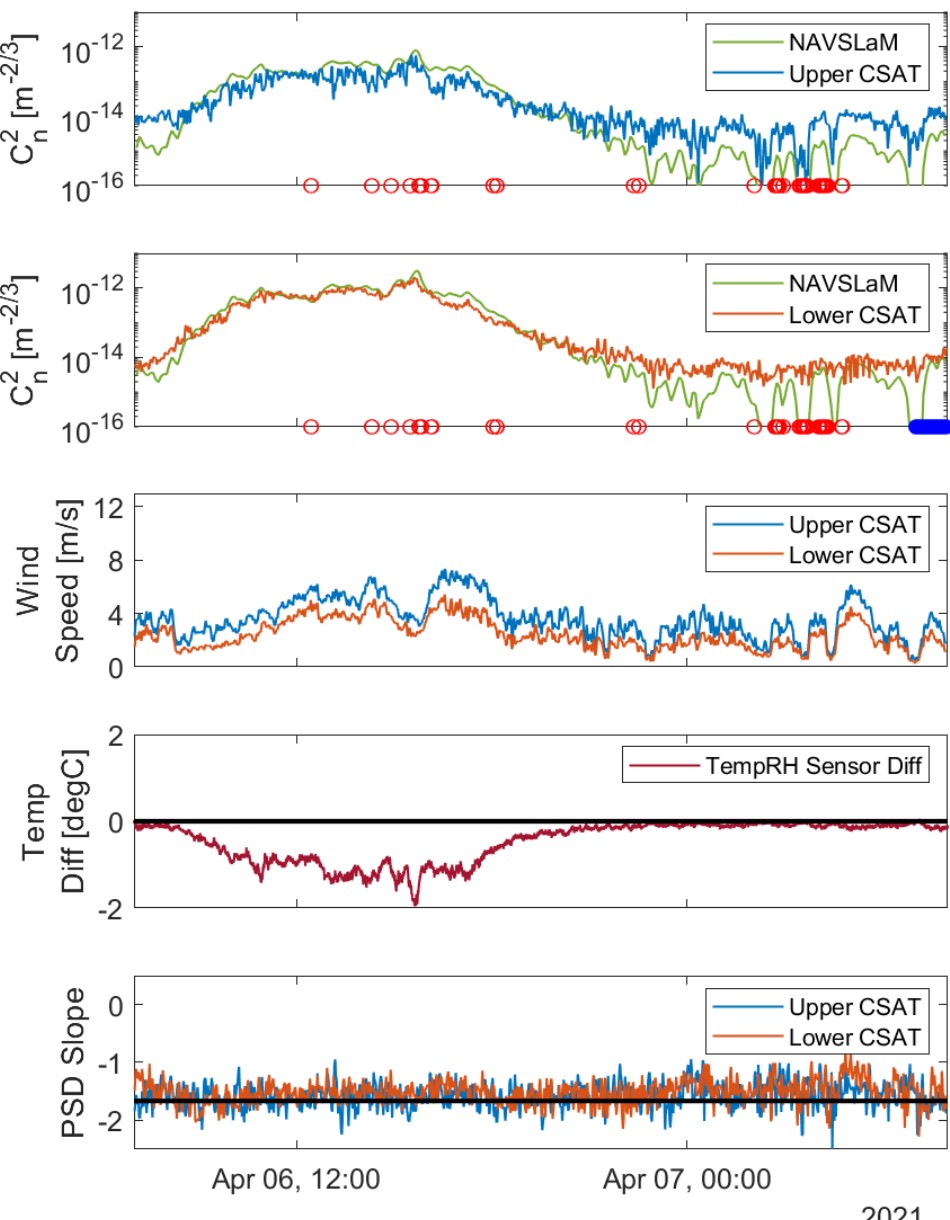

**Figure 5.** Comparison between $C_n^2$ derived from the CSAT3 data to NAVSLaM derived estimates, from 0700 Tuesday, 6 April to 0800 Wednesday, 7 April. The wind speed recorded from each CSAT3, the temperature difference between the CSAT3 positions, and the slope of the PSD are also provided. The red and blue circles along the axes of the upper two plots indicate times when the sonics returned diagnostic errors, as discussed in the text.

In the first plot of Figure 5, we see $C_n^2$ values measured by the upper sonic anemometer ranging from $10^{-16}$ m$^{-2/3}$ to $10^{-12}$ m$^{-2/3}$ with NAVSLaM estimates following a similar trend between the hours of 0800 and 1800 on the 6th of April. However, after this period, we begin to see some disagreement between NAVSLaM estimates and measured data. While NAVSLaM still follows the general trend of the sonic measurements, it begins to underestimate the $C_n^2$ values from the sonics with estimates as low as $10^{-17}$ m$^{-2/3}$. During these periods of underestimation, we also note that the temperature difference between the two HygroVUE5 sensors is almost zero. We know from previous research that NAVSLaM tends to underestimate $C_n^2$ values when the mean air temperature difference approaches

zero, i.e., when the conditions are "near neutral" [1]. In contrast, "stable" conditions occur when the mean air temperature difference is positive and "unstable" when it is negative. Additionally, we see $C_n^2$ values of $10^{-16}$ m$^{-2/3}$ measured by the sonic during this period of disagreement. This is potentially due to the upper sonic approaching the noise floor or vibrations caused by gusts of wind contaminating the measurements with noise. This is also confirmed by looking at the last plot of Figure 5, where the PSD slope deviates from $-5/3$ during this same period.

The second plot from the top in Figure 5 displays better agreement between the lower sonic $C_n^2$ measurements and NAVSLaM estimates. Between the hours of 0800 and 1800 on the 6th of April, we observed lower wind speeds recorded by the lower sonic anemometer as it is better shielded from the passing gusts of wind. However, the lower sonic is not completely impervious to wind speed effects as we note a similar rapid change in $C_n^2$ values during the post-midnight hours. Similar to the top plot of Figure 5, we see the greatest disagreement between measured sonic $C_n^2$ values and NAVSLaM estimates when the mean air temperature difference approaches near-neutral conditions. In contrast, we see the best agreement between measured data and estimates when the mean temperature difference is less than zero, or when prevailing conditions are unstable.

## 4. Discussion

The previous figures present a strong relationship between the mean temperature difference, wind speed, and $C_n^2$ values for both measured data and NAVSLaM estimates. Again, this trend is expected from Monin–Obukhov similarity theory upon which NAVS-LaM is based upon and is confirmed from the actual turbulence results derived from the sonic measurements. Figure 6 shows the data points collected and estimated for the entire time period of the experiment and is provided to further explore this relationship. The top two plots of Figure 6 display the NAVSLaM $\log(C_n^2)$ estimates versus the mean air temperature difference between the upper and lower CSAT3 locations. The bottom two plots show the CSAT3 sonic anemometer $\log(C_n^2)$ values at each level versus the mean temperature difference. The color of the dots corresponds to the wind speed associated with each data point, as indicated by the color bar to the right of each plot. The two upper plots have $\log(C_n^2)$ estimates that range from $-17$ to $-11$ and are compared against the CSAT3 sonic measurements which range from $-16$ to $-12$. Again, we see that NAVSLaM tends to underestimate $\log(C_n^2)$ values relative to the sonics as the mean temperature difference approaches zero shown in the first two plots. Both the NAVSLaM estimates and CSAT3-derived values indicate that for unstable conditions, an increase in wind speed reduces $C_n^2$, while the opposite is true for stable conditions. However, these trends are more prominent in the NAVSLaM results. The largest of the $\log(C_n^2)$ values are seen at the lower levels displayed in the top left and bottom left plots. We can attribute this to the sensor's proximity to the surface. It is also interesting to note the two separate curves seen in stable conditions of the NAVSLaM estimates. Previous research has attributed this occurrence to affects that relative humidity have on $C_n^2$ values [1]. This was not investigated during this experiment, but allows the opportunity for future work to analyze this relationship more thoroughly.

To gain a better understanding of how well NAVSLaM $C_n^2$ estimates agree with the CSAT3 sonic anemometer measurements, we displayed the difference in these values against the mean temperature difference in Figure 7. The top plot correlates to the values of the upper sensors and the bottom plot to the lower sensors. A solid black line is added on both plots to indicate periods of time that NAVSLaM estimates had a perfect agreement with the sonic measurements. In the unstable region, or when the temperature difference is less than zero, we see that NAVSLaM estimates tend to slightly overestimate the $C_n^2$ values compared to the upper sonic and yet have near-perfect agreement with the lower sensors. As seen in the previous figures, NAVSLaM begins to underestimate these values as conditions approach near-neutral regions. These estimates start to increase once more as we enter stable conditions, or a positive difference in mean temperature

measurements. However, the best agreements are still seen when the temperature difference is less than zero.

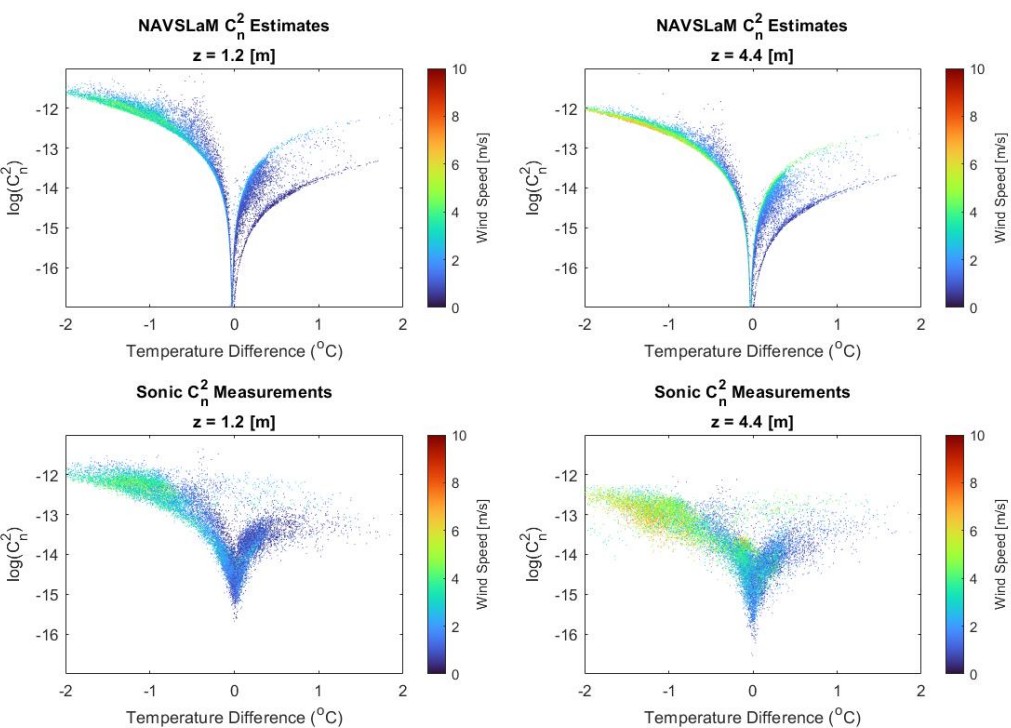

**Figure 6.** Summary plots for the entire experimental period, showing $log(C_n^2)$ from NAVSLaM estimates and CSAT3 sonic anemometer measurements versus the temperature difference between the upper and lower sonic locations. A temperature difference of zero indicates near-neutral conditions.

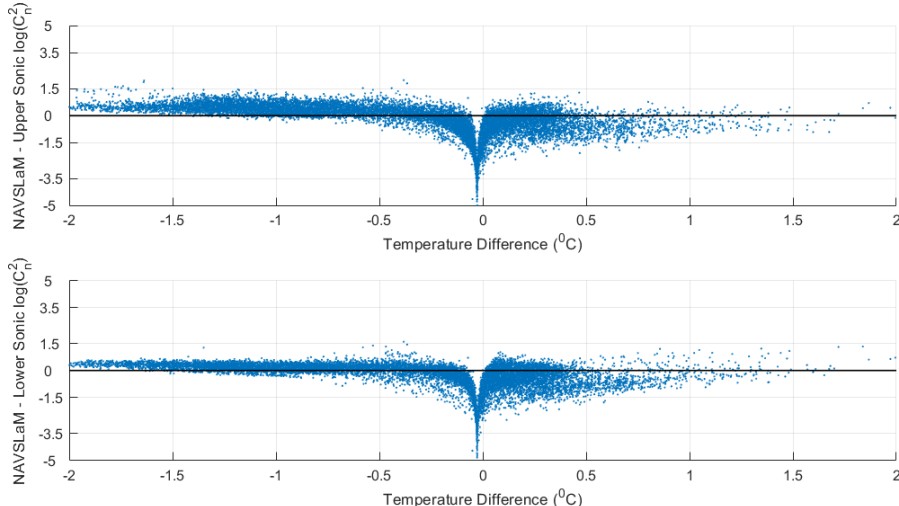

**Figure 7.** Difference between NAVSLaM-derived and CSAT3-derived $C_n^2$ values.

Although our results sometimes show significant differences between values of $C_n^2$ derived from sonic measurements and values predicted by NAVSLaM, it should be noted that when those differences occur, the values of $C_n^2$ tend to be very small. Such low levels of turbulence would have a smaller effect on laser beam propagation, so it is less essential to have accurate predictions of $C_n^2$ during those times.

## 5. Conclusions

These results provide validating evidence in support of the two-level version of NAVSLaM for a coastal environment. More specifically, when the mean temperature difference was negative, we observed the strongest values of optical turbulence as well as the greatest agreement between NAVSLaM estimates and measured data. These trends were especially evident during peak sunlight hours of the day when it was warmer closer to the ground, with $C_n^2$ values ranging from $\sim 10^{-13}$ m$^{-2/3}$ to $\sim 10^{-12}$ m$^{-2/3}$. We also observed the greatest disagreement between NAVSLaM and the measured data when the mean temperature difference approached zero. During these periods of disagreement, we were also careful to consider the sonic anemometer noise floor that might partially explain discrepancies during more quiescent conditions. However, there was generally good agreement between measured and predicted turbulence levels, except during near-neutral conditions, when the turbulence values tend to be smaller and thus would have less of an impact on laser beam propagation.

Due to COVID restrictions, the timeframe for collecting data for this experiment was limited to only about one month. In future work, we plan to collect data over a longer time in similar conditions to provide further validation of our results.

**Author Contributions:** Conceptualization, J.B. and K.C.; Formal analysis, S.B.; Investigation, S.B.; Methodology, S.B., J.B. and K.C.; Software, P.F. and K.C.; Validation, S.B. and P.F.; Writing—original draft, S.B., J.B., P.F. and K.C.; Writing—review and editing, J.B. and K.C. All authors have read and agreed to the published version of the manuscript.

**Funding:** This research was funded by Office of Naval Research, grant number N0001421WX01524.

**Institutional Review Board Statement:** Not applicable.

**Informed Consent Statement:** Not applicable.

**Data Availability Statement:** The data presented in this study are available on request from the corresponding author. The data are not publicly available due to a requirement for sponsor approval.

**Acknowledgments:** We wish to thank ONR for their support and Stephen Hammel for his advice and input.

**Conflicts of Interest:** The authors declare no conflict of interest.

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
