# Peer review of "Measurements and Modeling of Optical Turbulence in the Coastal Environment"

_applsci, doi:10.3390/app12104892_

Round 1

Reviewer 1 Report

This is a well organized measurement paper for inferring Cn2 by Ct2 measurement. The work is simple and well focused. The reviewer does not find distinctive issues in the work. 

Author Response

Thank you for taking the time to review our paper and we appreciate your feedback.

Reviewer 2 Report

Manuscript No:  applsci-1626498

Title:  Measurements and Modeling of Optical Turbulence in the Coastal Environment

Authors:  Sasha Barnett, Joseph Blau, Paul Frederickson, and Keith Cohn

  1. Overview
  2. In this manuscript the authors report on experimental work on Measurements and Modeling of Optical Turbulence in the Coastal Environment
  3. The contents are expressed clearly; the manuscript is reasonable organized and it is written in reasonable English.
  4. The authors have acknowledged recent related research.
  5. As long as my knowledge, the work presented is original and it is correct from a scientific point of view.

  1. Detailed analysis

Abstract: It is not clear for the reader. Please organize the ideas in each paragraph. Be clear, objective and self-explanatory. State briefly what you did, how did you do it, the quantitative results you and the novelty of your work. Please make it as synthetic as you can.

1.Introduction: provides an interesting approach to the subject but it could be a bit more informative an provide more up to date references.

  1. Materials and Methods: it provides a clear and correct explanation of the measurement structure.

  1. The authors must explain why the measurements were performed in a very limited time frame, from 16 March 2021 to 20 April 2021 as many variable may play a role.

Also gaps in the record due to power interruptions do not help in getting accurate results.

It is my understanding that the experiments should be repeat a few times and a statistical treatment would help reliable conclusions;

  1. Overall assessment

In my opinion the work should be reassessed after major changes.

  1. Review Criteria
  2. Scope of Journal

Rating: Medium

  1. Novelty and Impact

Rating: Medium

  1. Technical Content

Rating: Medium

  1. Presentation Quality

Rating: Medium

Author Response

Thank your taking the time to review our paper and we appreciate your feedback. Our specific replies to your comments and suggestions are in the attached document.

Reviewer 3 Report

The authors reported on the optical turbulence in the near-coastal environment of California, which was then compared with the refractive index structure parameter produced by a new version of the Navy Vertical Surface Layer Model (NAVSLaM). The measurements from the surface layer near the coastal area are valuable for the development of boundary layer theory and model.  Overall, this topic is worthy of investigation. Nevertheless, this manuscript can be accepted after fully addressing the following comments:

Introduction:

  1. First paragraph: The authors can review recent advances in the observation of optical turbulence from high-resolution radiosonde, which is highly relevant to the topic investigated here. Some of the recent publications are given below for authors’ reference:

https://doi.org/10.1029/2018JD029982

https://doi.org/10.1029/2019JD030287

https://doi.org/10.1088/1748-9326/abf461

  1. To my knowledge, except for the 3D sonic anemometer, there are quite a few other instruments that can be provide measurements of refractive index structure parameter. Unfortunately, I did not see any introductions in this regard. The authors are recommended to provide some background information in this regard, and to compare and discuss the instruments or measurements.
  2. “based upon based on” is redundant and should be reworded.
  3. The motivation of this manuscript is not clearly structured. Generally, the motivation for this study appears in the Introduction part. Therefore, the first paragraph in 2nd section can serve as motivation and thus should be moved to the end of introduction.

Results:

  1. Figure 2: Minor tick can be added at one-day intervals, from which the readers can see the diurnal variation or day-by-day variation.
  2. The authors are suggested to provide reasonable explanation for the phenomenon that the temperatures at the higher elevation were warmer than temperatures collected closer to the surface at the end of March. The authors can make an inlet figure showing a close-up view for this specific timespan. In this way, we can better figure out the possible physical explanation (warm air advection? sea breaze? Or something else?)

References:

  1. Ref #1 looks same as Ref. #9, and one should be removed from the reference list.

Round 2

Reviewer 2 Report

Manuscript No:  applsci-1626498 R1

Title:  Measurements and Modeling of Optical Turbulence in the Coastal Environment

Authors:  Sasha Barnett, Joseph Blau, Paul Frederickson, and Keith Cohn

  1. Overview
  2. In this manuscript the authors report on experimental work on Measurements and Modeling of Optical Turbulence in the Coastal Environment
  3. The contents are expressed clearly; the manuscript is reasonable organized and it is written in reasonable English.
  4. The authors have acknowledged recent related research.
  5. As long as my knowledge, the work presented is original and it is correct from a scientific point of view.

  1. In my opinion the work can be accepted for publication given that the authors made changes in the manuscript and

improved it.

  1. Review Criteria
  2. Scope of Journal

Rating: Medium

  1. Novelty and Impact

Rating: Medium

  1. Technical Content

Rating: Medium

  1. Presentation Quality

Rating: Medium